# Effect of Dietary Modulation of Selenium Form and Level on Performance, Tissue Retention, Quality of Frozen Stored Meat and Gene Expression of Antioxidant Status in Ross Broiler Chickens

**DOI:** 10.3390/ani9060342

**Published:** 2019-06-11

**Authors:** Doaa Ibrahim, Asmaa T.Y. Kishawy, Safaa I. Khater, Ahmed Hamed Arisha, Haiam A. Mohammed, Ahmed Shaban Abdelaziz, Ghada I. Abd El-Rahman, Mohamed Tharwat Elabbasy

**Affiliations:** 1Department of Nutrition and Clinical Nutrition, Faculty of Veterinary Medicine, Zagazig University, Zagazig 44519, Egypt; 2Department of Biochemistry, Faculty of Veterinary Medicine, Zagazig University, Zagazig 44519, Egypt; safaa_khater83@yahoo.com; 3Department of Physiology, Faculty of Veterinary Medicine, Zagazig University, Zagazig 44519, Egypt; vetahmedhamed@zu.edu.eg (A.H.A.); dr.haiam@yahoo.com (H.A.M.); 4Department of Pharmacology, Faculty of Veterinary Medicine, Zagazig University, Zagazig 44519, Egypt; asabdelaziz@vet.zu.edu.eg; 5Department of Clinical Pathology, Faculty of Veterinary Medicine, Zagazig University, Zagazig 44519, Egypt; gana660@gmail.com; 6College of Public Health and Molecular Diagnostics and Personalized Therapeutics Center (CMDPT) Hail University, Hail 2440, Saudi Arabia; tharwat330@gmail.com; 7Food Control Department, Faculty of Veterinary Medicine, Zagazig University, Zagazig 44519, Egypt

**Keywords:** broilers, selenium sources-levels, selenium retention, antioxidant capacity, frozen meat

## Abstract

**Simple Summary:**

Although the importance of usage of selenium as essential trace element in poultry production has been proven, the best source and level has not been fully addressed yet. Three different dietary selenium forms with three different levels were chosen to be added in broiler diet. Met-Se or nano-Se up to 0.6 mg/kg increased their performance and was more efficiently retained in the body than SeS. Frozen stored meat quality was improved in a dose-dependent manner especially with both Met-Se and nano-Se. Nano-Se was more potent than Met-Se, which in turn was more potent than inorganic Se against oxidative stress, which improved the quality of meat under frozen conditions.

**Abstract:**

This study compares between different selenium forms (sodium selenite; SeS, selenomethionine; Met-Se or nano-Se) and levels on growth performance, Se retention, antioxidative potential of fresh and frozen meat, and genes related to oxidative stress in Ross broilers. Birds (n = 450) were randomly divided into nine experimental groups with five replicates in each and were fed diets supplemented with 0.3, 0.45, and 0.6 mg Se/kg as (SeS, Met-Se), or nano-Se. For overall growth performance, dietary inclusion of Met-Se or nano-Se significantly increased (*p* < 0.05) body weight gain and improved the feed conversion ratio of Ross broiler chicks at the level of 0.45 and 0.6 mg/kg when compared with the group fed the same level of SeS. Se sources and levels significantly affected (*p* < 0.05) its concentrations in breast muscle, liver, and serum. Moreover, Se retention in muscle was higher (*p* < 0.05) after feeding of broiler chicks on a diet supplemented with Met-Se or nano-Se compared to the SeS group, especially at 0.6 mg/kg. Additionally, higher dietary levels from Met-Se or nano-Se significantly reduced oxidative changes in breast and thigh meat in the fresh state and after a four-week storage period and increased muscular pH after 24 h of slaughter. Also, broiler’s meat in the Met-Se and nano-Se groups showed cooking loss and lower drip compared to the SeS group (*p* < 0.05). In the liver, the mRNA expression levels of glutathione peroxidase, superoxide dismutase, and catalase were elevated by increasing dietary Se levels from Met-Se and nano-Se groups up to 0.6 mg/kg when compared with SeS. Therefore, dietary supplementation with 0.6 mg/kg Met-Se and nano-Se improved growth performance and were more efficiently retained than with SeS. Both sources of selenium (Met-Se and nano-Se) downregulated the oxidation processes of meat during the first four weeks of frozen storage, especially in thigh meat, compared with an inorganic source. Finally, dietary supplementation of Met-Se and nano-Se produced acceptable Se levels in chicken meat offered for consumers.

## 1. Introduction

Selenium (Se) is an important trace nutrient for the maintenance, growth, and animals and humans health [1]. It improves the nutritive value and of meat quality [2]. As feed additives, Se can enhance growth productivity in broiler chickens [3]. Selenium is an important constituent of at least 25 selenoproteins involved in various physiological processes, including immune function, reproduction, and the maintenance of antioxidant defenses to avoid tissue damage. Selenium deficiency results in a number of disorders and injuries in poultry, such as skeletal myodegeneration, exudative diathesis (ED), muscular hemorrhages, atrophy of pancreas, decreased production of eggs, liver injury, reduced hatchability, and inhibited growth of bursal and thymic [4], and increase susceptibility of humans to certain degenerative diseases, such as cancer [5]. The fortification of poultry meat with Se represents a viable strategy for increasing human intake of Se. The national research council (NRC) [6] recommendations established a low selenium level (0.15 mg/kg) for the supplementation of broilers. This level is not adequate to avoid production losses resulted from selenium deficiency disorders [7]; consequently, there is a need to increase dietary selenium levels. Moreover, Se bioavailability not only depends on its physical form but also on dietary concentration and the levels of other trace elements. Excess levels of Se can be toxic when provided above the biological requirement. Thus, meeting Se requirements and optimizing performance is an important step in modern poultry production. Practically, selenium can be added for poultry’s diet in the form of inorganic Se, organic Se, and most recently, nano-Se. The inorganic form of selenium (Se selenite) is primarily and commonly used for dietary supplementation, and exhibits a very narrow border between its dietary requirement and its toxicity [8]. Recently, it has been recognized that organic Se has a higher rate of tissue retention and bioavailability thus lower toxicity than inorganic Se, so it is preferable to inorganic Se in broilers [9]. In addition, organic Se is deposited more efficiently in breast muscle than inorganic forms [10]. With the development of nanotechnology, nano-Se has attracted widespread research interest due to its high catalytic efficiency and higher adsorbing capacity, and has exhibited strong absorption efficiencies and lower toxicity than inorganic Se [11]. Moreover, recent studies found that nano-Se has a higher effectiveness in controlling selenoenzymes and displays less toxicity than selenium-selenite [12]. Moreover, supplementation of dietary Se could also enhance oxide dismutase (SOD), glutathione peroxidase, (GPx) and catalase (CAT) activities, and reduce oxidative stress and lipid peroxidation biomarkers, consequently reducing oxidative stress in broilers [13]. Moreover, Se plays a key role in the signaling of redox via removal of hydrogen peroxide and lipid hydroperoxides using glutathione as an ultimate electron donor [14]. These antioxidant properties of Se have also been shown to continue in postmortem muscle tissue and prevent lipid oxidation [15]. For this reason, many dietary regimes in animal nutrition have been established to produce Se-enriched meat in order to increase human Se consumption [16]. The type and level of available Se is important to meet broilers’ dietary requirements and optimize their production without producing any hazardous effects on broilers or human health. Definitive comparative studies to fully exploit the benefits of dietary supplementation with different available Se sources and levels in Ross broilers remains poorly investigated. Thus, the aim of the present study was to compare the bioavailability of different levels and sources of Se on performance, Se retention, lipid oxidative stability of meat, meat quality, and mRNA expression of some selected genes related to antioxidant capacity in Ross broiler chickens.

## 2. Materials and Methods

### 2.1. Selenium Sources

Sodium selenite (SS) and selenomethionine (Met-Se) were obtained from Sigma-Aldrich Chemical Co., St. Louis, MO, USA and Sel-Plex; Alltech Inc., Nicholasville, KY, USA, respectively. nano-Se powder was prepared according to [17] where 4 mL of 25 mM GSH containing 15 mg of bovine serum albumin were mixed with one ml of 25 mM sodium selenite. The resulting red suspension was dialyzed against double-distilled water for 96 h. Every 24 h, the water was replaced to isolate the oxidized GSH from the nano-Se. Then, nano-Se and bovine serum albumin were lyophilized. The phase characterization and morphology of nanoparticles were analyzed by means of X-ray diffraction (XRD) using EMPYREAN diffractometer and JEM-200CX transmission electron microscopy (TEM) working at 30 kV as shown in (Figure 1). XRD patterns corresponding to the (100), (101), (110), (102), (111), (201), (003), (202), (210), and (211) planes of the formed nano-Se were observed at 20 angles of 23.6°, 29.9°, 41.4°, 43.8°, 45.4°, 51.8°, 55.9°, 61.8°, 65.3°, and 68.3°, respectively [18]. The nano-Se was spherical in shape with average size 42 ± 1.4 nm (total count of 100 NPs).

### 2.2. Birds and Experimental Procedures

Four hundred and fifty, one-day-old, Ross broiler chicks (Ross 308) were individually weighed and divided to nine dietary groups, each group consisting of five replicates of ten chicks each per floor pen. Broiler chicks were fed diets containing inorganic Se (sodium selenite; SeS), organic Se, (selenomethionine, Met-Se), or nano-Se, each at three levels 0.3, 0.45, or 0.6 mg/kg Se (as fed). The basal diet was formulated to meet nutrient requirements of Ross broilers according to [19] except Se (Table 1). Diets were fed from 1 to 38 d including starter (1–11 d), grower (12–22 d), and finisher (23–38 d) diets. All chicks were given ad libitum access to feed and water. The environmental temperature was kept at 32 °C for the 1st week and then gradually decreased until reached 23 °C. All the experimental procedures were performed at the Institute of Nutrition and Clinical Nutrition and Poultry Farm following the Faculty of Veterinary Medicine guidelines and in accordance with the protocols approved by Institutional Animal Care and Use committee at Zagazig University (Approval no: ZU-IACUC/2/F/123/2018).

The proximate analysis of feed ingredients was done according to the standard method of [20]. For Se analysis in feed, one gram of feed was heated for 5 h in a furnace at 550 °C for ashing. Mixture from 3 N HCl (10 mL) and an ashed sample was heated until the solution became clear. After cooling, the sample was filtered and diluted to 50 mL with 0.1 N HCl. For analysis of selenium, lanthanum 185.4 L 50 gm/kg was added to 6 mL of the sample solution. Then, analysis was achieved by a spectrometer at a wavelength of 400 nm [20].

### 2.3. Growth Parameter Measurement

Live body weight (LBW) and feed intake of broiler chicks/pen were estimated individually at 21 and 38 d of age to calculate live body weight, body weight gain (BWG), total feed intake, feed conversion ratio (FCR) and relative growth rate (RGR).

### 2.4. Sampling and Analytical Procedures

At the end of the feeding trail, tissues samples (liver and breast meat) were collected from five birds/replicates that were slaughtered (slaughtering house under supervision of Institutional Animal Care and Use Committee at Zagazig University Faculty of Veterinary Medicine) and handled and kept at −20 °C until analysis of selenium content and meat quality tests. Blood samples were collected with or without anticoagulant, then plasma and serum were kept at −20 °C until the analysis of selenium content and chemical analysis was performed.

#### 2.4.1. Tissue Retention of Selenium

Briefly, liver and breast muscle were weighed (0.1 g) and mixed with of HNO_3_ (8 mL) then digested by microwave. After that, deionized water was added to produce a 10 mL volume. The selenium content was determined following the procedure of [21] by atomic absorption spectrophotometer (Shimadzu Ltd., Shimane Shimadzu, Japan).

#### 2.4.2. Selenium Content in Serum Constituents

Selenium content was measured in serum by atomic absorption spectrophotometer (AA6501, Shimadzu Ltd., Japan). Plasma samples were used for measuring of aspartate amino transferase (AST), alanine glutamyl transferase (ALT), and creatinine calorimetrically by diagnostic kits (MAK055, MAK052, and C4255, respectively) manufactured by Sigma-Aldrich.

#### 2.4.3. Laboratory Analysis for Meat Quality

##### Meat pH and Drip and Cooking Loss in Meat Samples

Breast meat was used to determine postmortem pH (t = 0.5 and 24 h) by pH meter. Drip loss was estimated according to [22] (percent; proportional weight loss of a sample suspended for 72 h in a closed plastic bag under refrigerated conditions at 4 °C). After storage at −20 °C, cooking loss was determined (percent; weight loss proportionate of a sample after cooking for 40 min in a water bath at 70 °C followed by cooling).

##### Preparation of Samples for Total Antioxidant Capacity

Six hours after slaughter and handling, breast meat was cut into cubes of approximately 3 cm square); visible connective tissues and fat were removed. These muscle cubes mixed with distilled water then homogenized and centrifuged and used for measuring total antioxidative markers as free radical scavenging assay using 2,2′-azino-bis-3-ethylbenzothiazoline-6-sulfonic acid (ABTS) thiobarbituric acid reactive substances (TBARS) assay, and 2,2-diphenyl-1-picrylhydrazyl (DPPH) assay, and ferric reducing/ antioxidant (FRAP) assay.

Thiobarbituric acid-reactive assay: Oxidation was evaluated on the first day and after one week from storage by the thiobarbituric acid assay described by [23]. Perchloric acid (27 mL, 3.83% v/v) was added to of meat sample (5 g) then homogenized for 1 min and filtered by filter paper, then 2 mL thiobarbituric acid was added to supernatants and incubated in a water bath (100 °C) for 20 min. Subsequently, immediate cooling to room temperature and centrifugation for 15 min was performed, then the absorbance was read by the spectrophotometer at 532 nm. The results were then calculated according to the standard curve and values were expressed as mg malondialdehyde (MDA)/kg meat.

ABTS assay: The total antioxidant capacity of chicken breast and thigh meat was analyzed by Trolox-equivalent antioxidant capacity (TAC) assay [24]. Briefly, the reaction between 14 mM ABTS [2,2-azinobis-(3-ethylbenzothiazoline-6-sulfonic acid)] with an equal volume of 4.9 mM potassium persulfate was catalyzed to stimulate the formation ABTS+ radical cation formation, then incubated in the dark at room temperature for 12–16 h. After that, 10 μL of meat homogenate was added to the ABTS+ solution (1.0 Ml) and mixed thoroughly and after 60 s absorbance was read at 734 nm.

DPPH assay: The scavenging activity of the muscle samples was analyzed by 1,1-diphenyl-2-picrylhydrazyl radical (DPPH) [25]. Briefly, the meat samples were homogenized in distilled water and then centrifuged. The supernatant was mixed with ethanol and DPPH radical solution and incubated in a dark room for 10 min. Next, the absorbance measurement was read at 517 nm. The ability to scavenge the DPPH radical was expressed as μM per g of wet muscle tissue.

FRAP assay: Ferric reducing antioxidant power (FRAP) assay [26] was carried out on meat homogenates. The meat samples were homogenized in potassium phosphate buffer, centrifuged, and the supernatant was collected. Then, supernatant (1 mL) was collected and added to FRAP buffer (3 mL) containing 10 mM 2,4,6-Tris(2-pyridyl)-s-triazine) in 40 mM HCl, and 20 mM Fe_2_Cl_3_ was added to 300 mM acetate buffer. Immediately after mixing, the absorbance was measured at 593 nm. A standard curve was prepared with FeCl_2_. The antioxidant power of the samples was expressed as μM of Fe^2+^ per 1 g wet muscle tissue.

### 2.5. RNA Extraction, Reverse Transcription, and Quantitative Real-Time PCR.

At the end of the feeding trial (day 38), three birds per group were randomly selected, marked and injected with tert-butyl hydroperoxide, 0.2 mmol/kg body weight, intraperitoneally purchased from Sigma-Aldrich Chemical Co (St. Louis, MO, USA, CAS Number 75- 91-2) to induce the oxidative stress. Birds were slaughtered, and liver samples were collected 48 h post-injection. From liver tissue, the total RNA was extracted by RNeasy Mini Kit; Qiagen, Cat. No. 74104. according to the manufacturer regulation. The extracted RNA was quantified using the NanoDrop^®^ ND-1000 Spectrophotometer (NanoDrop Technologies, Wilmington, NC, USA). The first strand cDNA was synthesized using kits of RevertAid^TM^ H Minus (Fermentas Life Science, Pittsburgh, PA, USA). One μL of this cDNA was blended with 12.5 μL of 2× SYBR^®^ Green PCR mix with ROX from BioRad, 5.5 μL of RNase free water, and 0.5 μL (10 pmol/μL) of each forward and reverse primer for the selected genes were added. The primers’ sequences of catalase, glutathione peroxidase, and superoxide dismutase genes involved in antioxidant function were designed as previously described in [27]. The real-time PCR amplification was carried out with Rotor-Gene Q2 plex (Qiagen Inc., Valencia, CA, USA) with the following conditions; initial denaturation at 95 °C for 10 min and 40 cycles at 95 °C for 15 s and 60 °C for 1 min. Relative fold changes in the expression of target genes measured in triplicate were estimated by the comparative 2−ΔΔCt method with the GAPDH gene as an internal control to normalize target gene expression levels [28].

### 2.6. Statistical Analysis

Data were submitted to a 2-way ANOVA, using PASW statistics 18 (SPSS Inc., Chicago, IL, USA) to clarify the effects of dietary Se sources, its levels, and their interaction. Gene expression data were statistically analyzed using one-way ANOVA and relevant figures were generated by Graphpad Prism 7 (GraphPad Software Inc., San Diego, CA, USA). Tukey’s test was used to separate the means when the treatment difference was significant (*p* < 0.05). All data were expressed as the mean ± SEM. Statistical significance was considered at *p* ≤ 0.05.

## 3. Results and Discussion

### 3.1. Growth Performance

The effects of different dietary treatments on overall growth performance parameters (1–38 days) are presented in Table 2. The present study showed that the interaction between different dietary sources and levels of Se had significant (*p* < 0.05) effect on the body weight and gain of broilers at 38 days. The groups supplemented with selenomethionine (Se-Met) and nano-Se showed a significant increase (*p* < 0.05) in body weight and gain of Ross broiler chicks when compared with selenite selenium (SeS). Moreover, variety of levels and sources of Se played an important role in our study as when Se- Met or nano-Se were added to diets, body weight and gain increased as dietary Se levels increased, while higher levels in the SeS group at Se concentration of 0.3–0.45 mg/kg diet caused declines as dietary SeS levels increased. Different sources and levels of Se had no effect on feed intake (*p* < 0.05). Feed conversion ratio (FCR) of broilers was affected by the Se sources and levels as they play an important role in improving FCR. The FCR was improved by dietary supplementation of Se-Met or nano-Se, while the best FCR was established with a level of 0.6 mg Se/kg diet from Se-Met or nano-Se, followed by groups supplemented with 0.3–0.45 mg/kg from Se-Met or nano-Se. Our data demonstrated that the application of dietary Se-Met or nano-Se up to 0.6 mg/kg resulted in the maximum growth rate of broiler chicks, while the same dose from SeS tended to reduce the growth performance of broilers chicks. These results proved that such selected dietary Se-Met or nano-Se levels had higher bioavailability than inorganic forms of Se. These results are in agreement with [29] who found a decline in body weight gain and feed utilization as supplemental inorganic Se increased, while for nano-Se, average daily gain, FCR, and survival ratio reached their highest levels at an Se concentration of 0.15–1.20 mg/kg. [13] Showed that feeding of broilers on 0.3 and 0.5 mg/kg from nano-Se significantly improved FCR and increased tissue selenium content. Our findings were also in agreement with those of [30], who described no differences in feed intake among broilers fed diets supplemented with either organic or inorganic forms of Se. Broilers chicks groups fed on 0.2 mg/kg diet from organic selenium or nano-Se had a similar growth rate as compared to the group supplemented with the same level of Se-selenite [31]. Our findings were also in agreement with those of [32] who reported that increased selenium levels had improved average daily gain in the same time there was no differences on average daily gain between nano-selenium and organic selenium in broiler chickens. The function of Se on growth rate may relate to its role in the selenoprotein P and selenoenzymes type I iodothyronine deiodinase expression, which have critical roles in the synthesis of thyroid hormones and Se transport [33]. Moreover, our results of increased growth performance with selenium methionine and nano-Se could possibly due to an increased thyroid hormone regulating the body’s energy metabolism and increased digestibility of protein [34]. The results of this study suggest that different Se sources and levels may be necessary to optimize the performance of broilers, and that the form of organic Se may be of importance.

### 3.2. Selenium Retention in Serum, Muscle, and Liver

In the present study, the different Se sources and levels had significantly affected (*p* < 0.05) Se concentrations in serum, liver, and breast muscle of broilers (Table 3). The groups fed on a diet supplemented with nano-Se and Met-Se showed higher (*p* < 0.05) serum, liver, and breast muscle Se concentrations when compared with those fed diets supplemented with SeS, indicating that nano-Se and Se-Met were better retained in the body than SeS, although the effect of Se-Met was more prominent for tissue Se retention than nano-Se at the same lowered level. Accumulation of minerals in tissues is considered an indicator for mineral utilization [35]. The concept of increasing Se content in human foods by altering dietary Se sources and level given to livestock is now of interest to nutritionists [36]. Wang et al. [37] stated that transport and uptake of selenium by broiler intestinal cells were higher in nano-Se than that of SeS. The difference in retention of Se between Se yeast and SeS or nano-Se may be clarified by the probable metabolic pathways and absorption process for Se from different Se sources [29]. The safe limit of Se in human food has been established at 2.0 mg/kg for the United States [38]. This level agreed with our results that up to 0.6 mg/kg of Se in broiler diets precipitates less than 1 mg/kg in meat with all sources of Se. Selenium uptake from Se-selenite occurs by passive diffusion contributing the poor availability of Se-selenite [39], and up to 50–75% of consumed Se-selenite is lost through urine. Another limitation of adding selenite to feed is the short period storage of Se in the animal’s body [40]. Our results of Se retention in tissue in accordance with those of [41], who demonstrated that broiler chicks fed on dietary organic Se had higher (*p* < 0.05) Se content in breast muscle and liver than those fed diets fortified by SeS. [36] also proved that the contents of Se in liver and muscles were affected by dietary Se supplementation, and retention of Se was increased when organic Se was supplemented as compared with inorganic Se. Cai et al. [13] stated that increasing dietary nano-Se increased the concentration of selenium in liver and muscle tissue (*p* < 0.01). An explanation for increased tissue content from nano-Se may be attributable to improved intestinal absorption of nano-Se due to smaller particle size and larger surface area [42]. SeS and nano-Se, on the other hand, are changed to the transitional selenide and then employed for synthesis of selenoprotein or methylated and after that excreted. However, Met-Se contains a large amount of selenomethionine. When recognized as a Se species, it can be altered to selenocysteine through the trans-selenation pathway and then lysed to selenide. So Met-Se might be simply utilized in the tissue than SeS or nano-Se [43]. Another property of Met-Se involves the chemical similarity between Met-Se and Met, which permits the body to use them interchangeably in protein synthesis as Met-tRNA cannot distinguish between Met and Met-Se, which makes it possible to build Se reserves in the body [2].

### 3.3. The Effect of Different Levels and Sources of Se on Selected Serum Parameters

The activity of liver enzymes including ALT and AST were not significantly affected by the interaction between different levels and sources of Se. The same trend was recorded for serum creatinine values (Table 3). Selim et al. [44] stated that activity of liver enzymes including ALT and AST were not significantly affected by addition of Zn-Se-Meth, P-Nano-Se, or L-nano-Se in broiler diets. Moreover, increasing the supplemental Se level from 0.3 to 0.45 ppm in broiler diets could not cause any significant difference in plasma creatinine level. In previous studies, [44] found that liver enzymes were not affected by adding different forms of Se (inorganic, organic, or nano) at levels up to 0.3 mg Se/kg diet.

### 3.4. Antioxidant Potential of Different Sources and Levels of Se

These data showed that the expression pattern of selected antioxidant-related genes (glutathione peroxidase, GPx, super oxide dismutase, SOD and catalase, CAT) in relation to different Se levels and sources was addressed in (Figure 2). The expression of GPx mRNA significantly increased in groups fed nano-Se at 0.6 mg/kg diet followed by groups supplemented by 0.3 and 0.45 mg/kg diet from Met-Se and nano-Se when compared with SeS with the same levels. The highest expression of SOD was observed in groups supplemented with 0.45 and 0.6 mg/kg diet from nano-Se followed by the group supplemented by a 0.3 mg/kg diet from nano-Se and groups supplemented by 0.45 and 0.6 mg/kg diet from Met-Se, when compared with SeS supplemented group. The mRNA expression of catalase significantly increased with an increasing level of nano-Se and Met-Se when compared with SeS. The antioxidant enzymes such as SOD, CAT, and GPx [45], and non-enzymatic constituents such as glutathione (GSH) [46], play an important role for keeping the animal health, and physiological antioxidant systems. Selenium is a cofactor in several selenoproteins and the antioxidant selenoenzymes as glutathione peroxidase (GPx), thus its functional role is associated with the Se concentration in tissues [47]. Xiao et al. [48] demonstrated that the supplementation of Se in the maternal diet significantly (*p* < 0.05) enhanced the activity of GPx, T-SOD, and CAT in heat stress treated chick embryos when compared with the basal diet, as the levels of GPx1 mRNA were significantly (*p* < 0.05) elevated by adding Se. This may be clarified by higher Se retention in maternal Met-Se treatment [49], which aids in the production of more selenoproteins to preserve chick embryos with a higher antioxidant level. Under heat stress, [35] reported that the addition of organic Se significantly improved GPx activities as compared with broilers fed with inorganic Se. [50] established that the highest GPx activity and lowest MDA content in blood and testis were attained in the treatment of 0.5 mg/kg, as the GPx enzymes were involved in scavenging toxic H_2_O_2_ [51]. In animal research, the activity of GPx enzymes and their expression genes in tissues were correlated with the concentration of Se added to feed [47]. This finding is also in accordance with [52], who described that Se deficiency caused the reduction of GPx mRNA levels in four GPx genes found in chicken livers. The superoxide dismutase (SOD) and CAT are important antioxidant enzymes for poultry. The superoxide anion is transformed to H_2_O_2_ by SOD [53], and CAT changes H_2_O_2_ into water [54], although Se is not a component of SOD and CAT. Our results also indicated that nano-Se and Met-Se increased the mRNA expression of these genes. Yuan. [49] showed that in broiler breeding experiments, hepatic GPx1 and TrxR1 mRNA levels in Met-Se groups were higher (*p* < 0.05) than that in the SeS group.

### 3.5. Effect of Different Se Sources and Levels on Meat Quality

The role of diets supplemented with different Se sources and levels on breast meat quality in broiler chickens are shown in Table 4. Compared with SeS, dietary Met-Se and nano-Se inclusion in broiler diet improved meat quality, especially as Se levels increased from 0.45 to 0.6 mg/kg.

### 3.6. Post-Mortem pH of Meat, Cooking Loss and Drip Loss

Breast meat from groups that received an increased level of Met-Se and nano-Se exhibited increased (*p* < 0.05) pH 0.5 and 24 h later when compared with the SeS groups. In addition, birds in the Met-Se and nano-Se groups, specially at high levels, had lesser drip and cooking loss group (*p* < 0.05) compared to those in the SeS groups. The presence of Se in animal diets are a key influence on meat water retention, with the form and level regulating the variation in meat drip loss [55]. The results of our study agreed with [56], who found that the drip loss was lower and water-holding capacity was higher in pigs fed with organic selenium. It has been reported by some authors that the mechanism by which antioxidants modify the water-holding capacity and drip loss of meat can be attributed to their ability to maintain muscle membranes’ integrity post-mortem [57], while others have suggested that proteolysis and protein oxidation acts as an essential factor for determining the moisture retention of meat [58]. Lambert et al. [59] reported that the accumulation of a large amount lactic acid in the muscles, combined with a cessation of blood circulation which induces cellular hypoxia and results in a decreased pH after slaughter, changed the permeability of cell membrane and decreased the water-holding capacity. But our study demonstrated increased water-holding capacity of breast meat in broilers fed on Met-Se and nano-Se. This may be explained by the metabolic conversion of glucose to lactic acid in post-mortem muscle being delayed with organic Se or nano-Se supplementation, thus improving the water-holding capacity of meat and decreasing drip loss [60]. It has been reported elsewhere that 0.3 mg/kg Met-Se supplementation resulted in an increase in the pH of the breast meat of broilers [61] and in geese [62] as compared with 0.3 mg/kg SS supplementation. Other studies demonstrated that water-holding capacity is affected by organic Se supplementation [63] and nano-Se [64]. Cai et al. [13] proved that application of nano-Se increases the ability of broiler muscle proteins to attract water, thus reducing drip loss percentage. The present study further indicates that the role of Met-Se and nano-Se on the biochemistry of muscle tissue is more prominent than with SeS.

### 3.7. Thiobarbituric acid Reactive Substances (TBRAS) Content of Meat as a Marker for Lipid Oxidation

Frozen storage of all analyzed meat significantly increased (*p* < 0.05) the TBRAS values in breast and thigh meat, with the lowest values for TBRAS recorded in breast meat, which could be related to the total lipid content. With increasing dietary level of organic Se and nano-Se, the TBRAS values decreased in breast and thigh meat when compared with SeS supplementation (Table 4). Exposure to different physiochemical or pathological conditions has recently been shown to be one of the main predisposing agents controlling free radical formation in the body [65]. On the other hand, chicken meat enriched with polyunsaturated fatty acids (PUFA) augmenting the meat susceptibility to oxidation progressions [66]. Bakhshalinejad [32] reported that oxidation resistance of broiler meat was higher in case of supplementation of organic of Se and the higher concentration of Se the higher glutathione peroxidase activity, total antioxidant capacity and malondialdehyde formation. Oxidation of lipids produces free radicals, leading to mutagenesis, carcinogenesis, and aging of the cell [67]. The antioxidant role of Se has also been shown to continue post-mortem in muscle tissue, where it is reported Se reduced oxidization of lipids in meat and had an effect on its quality [15]. Providing Se-enriched meat for human consumption by manipulating animal feed therefore also protects the quality of meat [68]. In this respect, [69] showed that inclusion of Se in poultry diets provides Se-enriched meat and protects the meat from oxidation after slaughter, increasing the stability of the meat against various storage conditions which accelerate the oxidation processes that destroy membrane lipids, consequently reducing the meat’s nutritional value [70]. Similarly, higher protection of muscle samples against lipid oxidation have been demonstrated by Se yeast with broilers [61] and turkey meat [71]. In addition, the breeders’ diet supplemented with Se also provides antioxidant protection of lipid rich tissues, which was detected by lower TBARS values after slaughter [72]. Calvo et al. [73] found that birds supplemented with organic Se had lower malondialdehyde (MDA) concentrations in muscle samples than the SeS group with the same storage time. In agreement with our results on muscle pH, it has been reported that the pH reduction could accelerate lipid oxidation due to the enhanced autoxidation of hemoglobin at reduced pH [74]. With decreasing muscle pH, higher TBARS values have been reported [75].

### 3.8. Total Antioxidant Capacity of Meat

The presence of antioxidants in poultry meat is a powerful factor influencing its quality. Once antioxidant defense systems are debilitated, dysfunction of all body cells and tissues may occur. Thus to keep body functions optimal, antioxidant levels are important [76]. As Se plays major role in protecting cells against oxidative stress, measuring the antioxidant biomarkers is a beneficial tool for evaluating the Se antioxidative role. In the present study ABTS, DPPH, and FRAP assays were used to estimate antioxidant capacities, as theses assay reflect the antioxidant properties of meat [77].

DPPH Assay: Thigh meat was characterized by significantly higher DPPH free radical scavenging ability than breast meat. The supplementation of nano-Se and organic Se at higher levels (0.6 mg/kg) into the Ross broiler diet increased the ability of meat to scavenge free radical DPPH and this capacity increased with the storage period (Table 4). Using specific sources from selenium in poultry diet increases the meat’s ability to scavenge the free radical DPPH, due to Se antioxidative functions. During frozen storage, the removal ability of DPPH augmented in all examined samples of chicken meat, demonstrating that Se is stable in the meat [78].

ABTS Assay: The ability of breast and thigh meat to scavenge free radical ABTS were affected by dietary inclusion of Met-Se and nano-Se up to 0.6 mg/kg. During frozen storage, the ability of the meat parts to remove the free radical ABTS tended to increase, reaching the highest values after four weeks of storage (Table 4). These results agree with [78], who stated a higher antioxidative potential of chickens breast to remove free radicals tended to increase during frozen storage, reaching the highest values after storage period of 90 days. This can be accompanied by moisture loss as a result of evaporation, besides alterations in proteins structure and lipids due to oxidation progressions. Also, implementation of Met-Se and nano-Se to chicken diets significantly rises the breast’s ability and thigh tissues to scavenge the synthetic free radical ABTS when compared with SeS.

FRAP Assay: In general, the capacity of the thigh myofibrillar protein to reduce Fe^3+^ to Fe^2+^ was higher than in breast myofibrillar protein. In the first three hours, dietary inclusion of 0.45 and 0.6 mg/kg diet of Met-Se and nano-Se had the same reducing capacity of Fe^3+^ to Fe^2+^, while after four weeks the reducing capacity of Fe^3+^ to Fe^2+^ was more prominent in breast meat and thigh meat for groups supplemented with nano-Se. It is well understood that Se is vital for the intra- and extra-cellular antioxidant systems of the body [79]. Selenium is also effective in delaying post-mortem oxidation responses [15]. The association between meat quality and oxidation resistance of muscle is well recognized. Huff-Lonergan et al. [58] described that changes in the antioxidant defense system of animals and muscles would affect water-holding capacity, meat proteolysis and calpain activity, thus quality characteristics of meat. In former studies, the water-holding capacity and chicken muscles color were enhanced by dietary Se addition [80]. Se application to chicken diets causes a significant increase in the iron reduction ability for both sets of the leg and back muscles, which can be associated with the higher Se retention in the lipids-rich parts [78]. Li et al. [81] described how total protein solubility, pH at 45 min, and myofibrillar protein solubility decreased while cooking loss was improved after feeding broiler chickens 0.3 mg/kg of either Met-Se or nano-Se as compared with SeS. Muscle proteins comprise connective tissue, sarcoplasmic and myofibrillar, [82]. Protein solubility resulted from protein denaturation during muscle ageing. In addition, denaturation of muscle protein is associated with antioxidant capacity [83]. When muscle amino acids as cysteine, tryptophan are oxidized, disulfide bonds and carbonyl are produced. At that time, the protein structure is destroyed, which would decrease the solubility of protein [84]. Current study, revealed significant increases in the iron reduction capacity which can be related to higher deposition of Met-Se and nano-Se in breast and thigh of chickens specially when supplemented with higher dose (0.6 mg /kg diet) compared with SeS supplementation, which could be a consequence of improved antioxidant capacity.

## 4. Conclusions

Our results suggested that in Ross broiler chickens, dietary supplementation of either Met-Se or nano-Se up to 0.6 mg/kg increased their performance and was more efficiently retained in the body than SeS. In addition, under stress the antioxidant resistance of broilers fed selected higher levels of Met-Se or nano-Se was enhanced. Moreover, frozen stored meat quality was improved in a dose-dependent manner with both Met-Se and nano-Se. Nano-Se was more potent than Met-Se, which in turn was more potent than inorganic Se against oxidative stress, which improved the quality of meat under frozen conditions.

## Figures and Tables

**Figure 1 animals-09-00342-f001:**
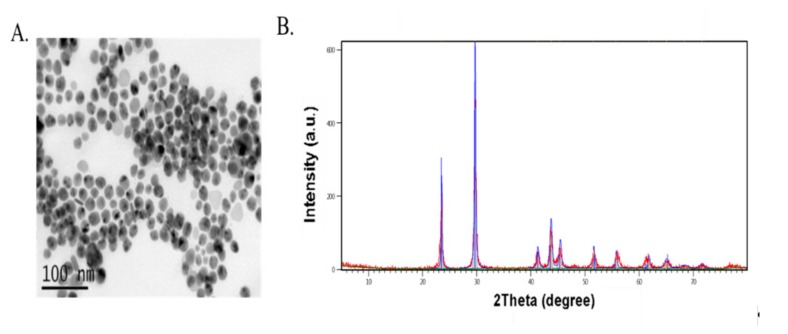
Characterization of nano-Se (**A**,**B**): (**A**) Morphology of the formed nano-Se pictured by transmission electron microscopy (TEM) and; (**B**) X-ray diffraction (XRD) pattern of the nano-Se.

**Figure 2 animals-09-00342-f002:**
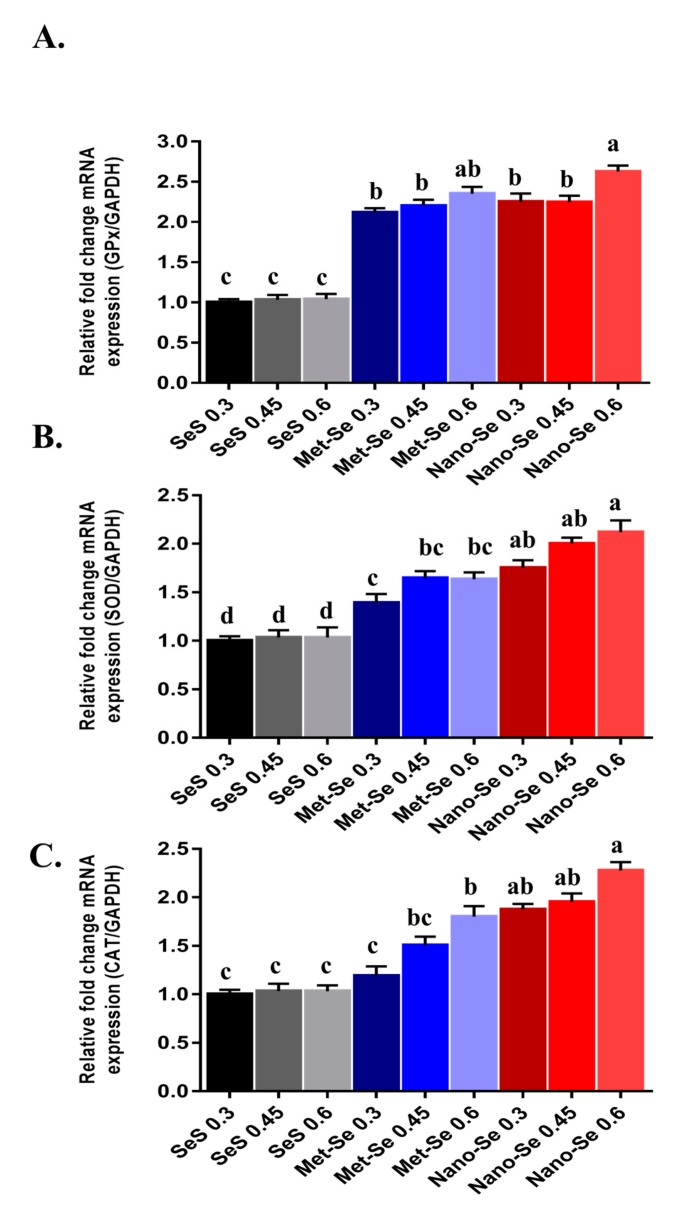
Effects of dietary Se source and level on the relative antioxidant enzymes expression (**A**–**C**). (**A**) Glutathione peroxidase (GPx); (**B**) super oxide dismutase (SOD); and (**C**) catalase (CAT) in the liver of broiler chickens at 40 days. SeS = sodium selenite, Met-Se = selenomethionine, nano-Se = nano-selenium. ^a,b,c,d^ Different superscript letters denote significant difference (*p* < 0.05). Values are means ± standard error.

**Table 1 animals-09-00342-t001:** The ingredients and nutrient levels of the basal diet (on dry matter basis).

Ingredients	Starter	Grower	Finisher
Corn, %	56	61.7	62.5
Soybean meal, %	34.86	28.1	25
Corn gluten, %	3.5	3.3	3.5
wheat bran, %	0	1	1.9
Soy oil, %	1.8	2.2	3.66
Calcium carbonate, %	1	1	1
Calcium diphasic phosphate, %	1.8	1.7	1.5
NaCl, %	0.3	0.3	0.3
Premix *, %	0.3	0.3	0.3
Methionine, %	0.18	0.14	0.11
Lysine, %	0.16	0.16	0.13
anti-mycotoxin, %	0.1	0.1	0.1
Total	100	100	100
Nutrient Levels ^b^			
Crude protein, %	23.2952	20.527	19.3087
ME (kcal/kg)	3042.271	3105.028	3200.17
Calcium, %	0.9656	0.92681	0.86886
Available P, %	0.467822	0.43785	0.3962
Methionine, %	0.569576	0.49246	0.456018
Lysine, %	1.380138	1.18469	1.092276
Ether extract, %	4.28232	4.8086	2.6345
Crude fiber, %	2.64082	2.6282	6.2493
Se mg/kg	0.06986	0.0696	0.07615

* Provided for each kilogram of diet: Vitamin A, 10,000 IU; vitamin E, 7200 IU; vitamin D3, 3000 IU; vitamin K, 2 mg; vitamin B1, 2640 mg; vitamin B6, 1200 mg; calcium pantothenate, 10 mg; nicotinamide, 50 mg; biotin, 40 mg; choline chloride, 500 mg; folic acid, 0.5 mg; cobalamin, 0.01 mg; calcium, 9000 mg; manganese, 120 mg phosphorus, 2100 mg; sodium, 3700 mg; iron, 110 mg; copper, 10 mg; zinc, 100 mg; iodine 1.1 mg. ^b^ Calculated values except selenium.

**Table 2 animals-09-00342-t002:** Effects of dietary sources and levels of Se (mg/kg) on growth performance of broilers over 38 days.

	SeS			Met-Se			Nano-Se			SEM	*p*-value		
	0.3	0.45	0.6	0.3	0.45	0.6	0.3	0.45	0.6		Se Source	Se Level	Source × Level
BW	2184 ^c^	2262 ^b^	2155 ^c^	2266 ^b^	2304 ^b^	2391 ^a^	2253 ^b^	2263 ^b^	2372 ^a^	4.82	<0.001	<0.001	<0.001
BWG	2139 ^c^	2217 ^b^	2110 ^c^	2219 ^b^	2260 ^b^	2346 ^a^	2211 ^b^	2214 ^b^	2327 ^a^	4.18	<0.001	<0.001	<0.001
FI	3932	3877	3881	3823	3879	3859	3811	3826	3913	12.09	0.227	0.585	0.215
FCR	1.84 ^c^	1.75 ^b^	1.86 ^c^	1.72 ^b^	1.71 ^b^	1.64 ^a^	1.73 ^b^	1.73 ^b^	1.68 ^a,b^	0.005	<0.001	0.007	<0.001
RGR	192.03	192.30	191.90	192.08	192.42	192.62	192.18	192.21	192.62	0.17	0.740	0.775	0.926

SeS = sodium selenite; Met-Se = selenomethionine; nano-Se = nano-selenium; BW = body weight; BWG = body weight gain; FI = feed intake; FCR = feed conversion ratio, RGR = relative growth rate. ^a,b,c^ Means within a row carrying different superscript letters denote significant differences (*p* < 0.05).

**Table 3 animals-09-00342-t003:** Effects of dietary Se sources and levels on Se concentration and some plasma biochemical parameters of broilers over 38 days ^1^.

	SeS			Met-Se			Nano-Se			SEM	*p*-Value		
	0.3	0.45	0.6	0.3	0.45	0.6	0.3	0.45	0.6		Se Source	Se Level	Se Source × Level
Se concentration													
Serum Se, mg/L	0.17 ^f^	0.32 ^e^	0.42 ^d^	0.20 ^f^	0.48 ^c^	0.64 ^a^	0.20 ^f^	0.54 ^b^	0.63 ^a^	0.005	<0.001	<0.001	<0.001
Liver Se, mg/kg	0.32 ^f^	0.94 ^d^	1.16 ^c^	0.52 ^e^	1.27 ^c^	1.52 ^b^	0.59 ^e^	1.55 ^b^	1.83 ^a^	0.011	<0.001	<0.001	<0.001
Muscle Se, mg/kg	0.16 ^g^	0.27 ^e^	0.32 ^c,d^	0.28 ^d,e^	0.41 ^b^	0.75 ^a^	0.22 ^f^	0.35 ^c^	0.77 ^a^	0.005	<0.001	<0.001	<0.001
Plasma biochemistry													
AST U/L	83.50	84.28	84.78	83.59	83.63	83.71	83.43	84.12	84.17	0.12	0.173	0.049	0.559
ALT U/L	152.91	152.94	155.06	152.81	152.84	152.91	152.81	152.59	153.18	0.028	0.175	0.097	0.320
Creatinine mg/dL	5.68 ^b^	5.84 ^a,b^	6.50 ^a^	5.70 ^a,b^	5.58 ^b^	6.04 ^a,b^	5.94 ^a,b^	5.92 ^a,b^	6.20 ^a,b^	0.009	0.162	0.002	0.521

SeS = sodium selenite; Met-Se = selenomethionine; nano-Se = nano-selenium. ^a,b,c,d,e,f^ Within a row, different superscript letters denote significant difference (*p* < 0.05). ^1^ Values are means ± standard error.

**Table 4 animals-09-00342-t004:** Effects of dietary Se source and level (mg/kg) on meat quality after slaughter and antioxidative potential of broiler meat (breast and thigh) during frozen storage.

	SeS			Met-Se			Nano-Se			SEM	*p*-Value		
	0.3	0.45	0.6	0.3	0.45	0.6	0.3	0.45	0.6		Se Source	Se Level	Se Source × Level
pH, 0.5 h	6.37 ^d,e^	6.49 ^c,d^	6.32 ^e^	6.56 ^c^	6.67 ^b^	6. 80 ^a^	6.78 ^a^	6.82 ^a^	6.83 ^a^	0.007	<0.001	<0.001	<0.001
pH, 24 h	5.46 ^d^	5.48 ^c,d^	5.41 ^d^	5.54 ^b,c^	5.56 ^b,c^	5.67 ^a^	5.59 ^b^	5.62 ^a,b^	5.70 ^a^	0.005	<0.001	<0.001	<0.001
Drip loss, %	2.74 ^a^	2.62 ^b^	2.83 ^a^	2.38 ^c^	2.17 ^d^	2.13 ^d,e^	2.35 ^c^	2.16^d^	2.04 ^e^	0.007	<0.001	<0.001	<0.001
Cooking loss %	14.04 ^b^	12.98 ^c^	14.60 ^a^	12.96 ^c^	12.64 ^c,d^	12.16 ^e^	12.86^c^	12.18 ^d,e^	12.07 ^e^	0.03	<0.001	<0.001	<0.001
Breast TBRAS, mg/kg 3 h	0.17 ^a^	0.17 ^a^	0.14 ^a^	0.14 ^a^	0.11 ^a,b^	0.04 ^c^	0.11 ^a^	0.05 ^b,c^	0.03^c^	0.02	<0.001	<0.001	<0.038
Breast TBRAS, mg/kg 2 W	0.47 ^a^	0.38 ^b,c^	0.42 ^a,b^	0.36 ^b,c^	0.30 ^c,d^	0.26 ^d^	0.39 ^a,b,c^	0.22 ^d,e^	0.15 ^e^	0.02	<0.001	<0.001	<0.001
Breast TBRAS, mg/kg 4 W	0.83 ^a^	0.78 ^a,b^	0.77 ^a,b^	0.71 ^b,c^	0.71 ^b,c^	0.68 ^c,d^	0.64 ^d^	0.53 ^e^	0.50 ^e^	0.02	<0.001	<0.001	<0.007
Thigh TBRAS, mg/kg 3 h	0.23 ^a^	0.26 ^a^	0.24 ^a^	0.25 ^a^	0.21 ^a,b^	0.13 ^c^	0.23 ^a^	0.16 ^b,c^	0.11^c^	0.02	<0.001	<0.001	<0.001
Thigh TBRAS, mg/kg 2 W	0.48 ^a^	0.40 ^b^	0.47 ^a^	0.35 ^b,c^	0.29 ^c,d^	0.31 ^c^	0.39 ^b^	0.23 ^d,e^	0.18 ^e^	0.02	<0.001	<0.001	<0.001
Thigh TBRAS, mg/kg 4 W	0.86 ^a^	0.80 ^a^	0.80 ^a^	0.72 ^b^	0.72 ^b^	0.69 ^b,c^	0.65 ^c^	0.56 ^d^	0.54^d^	0.02	<0.001	<0.001	<0.002
Breast ABTS, 3 h	2.30 ^e^	2.47 ^d^	2.09 ^f^	3.47 ^c^	3.60 ^c^	3.75 ^b^	3.87 ^b^	4.20 ^a^	4.21 ^a^	0.03	<0.001	<0.001	<0.001
Breast ABTS, 4 week	5.37 ^f^	5.59 ^e^	5.27 ^g^	6.31 ^d^	6.52 ^c^	6.93 ^b^	6.92 ^b^	8.14 ^a^	8.21 ^a^	0.02	<0.001	<0.001	<0.001
Thigh ABTS, 3 h	6.47 ^f^	6.73 ^d,e^	6.93 ^c^	6.70 ^e^	7.04 ^b,c^	6.95 ^c^	6.90 ^c,d^	7.18 ^b^	8.18 ^a^	0.05	<0.001	<0.001	<0.001
Thigh ABTS, 4 week	7.47 ^e^	8.06 ^b,c^	7.94 ^c^	7.70 ^d^	7.73 ^d^	7.95 ^c^	7.90 ^c^	8.18 ^a,b^	9.18 ^a^	0.04	<0.001	<0.001	<0.001
Breast DPPH, 3 h	5.91 ^e^	6.19 ^d^	5.81 ^f^	5.98 ^e^	6.29 ^c,d^	6.30 ^c,d^	6.34 ^c^	6.74 ^b^	7.13 ^a^	0.04	<0.001	<0.001	<0.001
Breast DPPH, 4 week	6.39 ^d^	6.71 ^c^	6.35 ^d^	6.48 ^d^	6.79 ^c^	6.80 ^c^	6.85 ^c^	7.34 ^b^	6.77 ^a^	0.05	<0.001	<0.001	<0.001
Thigh DPPH, 3 h	7.14 ^d,e^	7.48 ^a,b^	7.01 ^e^	7.21 ^c,d,e^	7.20 ^c,d,e^	7.36 ^b,c,d^	7.42 ^a,b,c^	7.45 ^a,b^	7.61 ^a^	0.05	<0.001	<0.03	<0.001
Thigh DPPH, 4 week	8.54 ^f^	8.82 ^e^	8.83 ^e^	9.23 ^d^	9.39 ^c^	9.58 ^b^	9.41 ^c^	9.58 ^b^	9.72 ^a^	0.02	<0.001	<0.001	<0.02
Breast FRAP, 3 h	0.15 ^d^	0.22 ^b,c^	0.16 ^c,d^	0.22 ^b,c^	0.32 ^a^	0.33 ^a^	0.25 ^b^	0.33 ^a^	0.37 ^a^	0.01	<0.001	<0.001	<0.001
Breast FRAP, 4 week	0.28 ^c^	0.38 ^b^	0.30 ^c^	0.37 ^b^	0.43 ^a,b^	0.42 ^a,b^	0.40 ^a,b^	0.43 ^a,b^	0.45 ^a^	0.01	<0.001	<0.001	<0.012
Thigh FRAP, 3 h	1.06 ^f,g^	1.13 ^e,f^	1.03 ^g^	1.15 ^e^	1.19 ^d,e^	1.35 ^a,b^	1.26 ^c,d^	1.30 ^b,c^	1.40 ^a^	0.02	<0.001	<0.01	<0.001
Thigh FRAP, 4 week	1.24 ^a^	1.28 ^a^	1.26 ^a^	1.31 ^a^	1.40 ^c^	1.52 ^a,b^	1.39 ^c^	1.50 ^b^	1.59 ^a^	0.02	<0.001	<0.001	<0.001

SeS = Sodium selenite; Met-Se = selenomethionine; nano-Se = nano-selenium; ^a,b,c,d,e,f,g^ means within a row carrying different superscript letters denote significant difference (*p* < 0.05).

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
