# Peer review of "Effect of Dietary Modulation of Selenium Form and Level on Performance, Tissue Retention, Quality of Frozen Stored Meat and Gene Expression of Antioxidant Status in Ross Broiler Chickens"

_animals, 2019, doi:10.3390/ani9060342_

Round 1

Reviewer 1 Report

Dear Editor,

thank you for the opportunity to review the paper “Animals-501351” entitled “Consequences of substituting dietary nano selenium for its other sources on growth, biomarkers of antioxidant capacity of frozen stored meat, and gene expression of antioxidant status in broilers.” This manuscript is interesting and focus the scientific attention on an important topic about dietary addition with nanoelements in meat production, particularly poultry meat.

The paper is well done, the experimental design is well structured, and the results and discussion are characterized by good scientific sound.

Just few minor remarks reported below:

Materials and Methods

Lines 121-123 it is not clear how many animals were involved. 400? After this you write about 9 dietary groups, and each group is formed by 5 replicates of 10 animals. So, the total is 450 animals. Probably there is a mistake because the first sentence leads the reader to understand that you involved 400 animals at the age of 51 days old. Please revise this.

What about slaughtering procedures? Where did you do them? You wrote (line 164) that 5 animals were slaughtered for blood sampling. One for each replication? And the other 45 for each feeding treatment? How many samples did you analyse for blood analysis and how many for meat quality? After (line 215) you wrote about 3 animals slaughtered for other analysis. For each feeding treatment or replication? And so, how many samples you consider for meat analysis. I think you need to be more precise.

Some references, if possible, can be “renewed”, you can consider more recent papers that apply the same method (for example n. 22, 23, 24 for meat analysis.)

Author Response

Dear: Dr Reviewer 1

All your comments were taken carefully and corrected in the manuscript 

Comments and Suggestions for Authors

Dear Editor,

thank you for the opportunity to review the paper “Animals-501351” entitled “Consequences of substituting dietary nano selenium for its other sources on growth, biomarkers of antioxidant capacity of frozen stored meat, and gene expression of antioxidant status in broilers.” This manuscript is interesting and focus the scientific attention on an important topic about dietary addition with nanoelements in meat production, particularly poultry meat.

The paper is well done, the experimental design is well structured, and the results and discussion are characterized by good scientific sound.

Just few minor remarks reported below:

Materials and Methods

Lines 121-123 it is not clear how many animals were involved. 400? After this you write about 9 dietary groups, and each group is formed by 5 replicates of 10 animals. So, the total is 450 animals. Probably there is a mistake because the first sentence leads the reader to understand that you involved 400 animals at the age of 51 days old. Please revise this.

Response: This part was revised and corrected.

What about slaughtering procedures? Where did you do them? You wrote (line 164) that 5 animals were slaughtered for blood sampling. One for each replication? And the other 45 for each feeding treatment? How many samples did you analyze for blood analysis and how many for meat quality? After (line 215) you wrote about 3 animals slaughtered for other analysis. For each feeding treatment or replication? And so, how many samples you consider for meat analysis. I think you need to be more precise.

Response: we cleared this part in sampling and analytical procedures line 155

Some references, if possible, can be “renewed”, you can consider more recent papers that apply the same method (for example n. 22, 23, 24 for meat analysis.)

 Response: references was revised and updated

Reviewer 2 Report

As stated in the Introduction, the aim of the present study was to compare the bioavailability of different levels and sources of Se on performance, Se retention, lipid oxidative stability of meat, meat quality, and mRNA expression of some selected genes related to antioxidant capacity in Ross broiler chickens.

It is however not difficult to find recent papers on the same topic with similar selenium sources in broiler chickens. One exception is that current study employed three methodologies to measure antioxidative activities in stored meat samples. In addition, there are many papers describing the functions of selenium in broiler chickens.

Bakhshalinejad et al. 2018. Effects of different dietary sources and levels of selenium supplements on growth performance, antioxidant status and immune parameters in Ross 308 broiler chickens, which published at British Poultry Science.

In Introduction, authors indicated that organic Se is preferable to inorganic Se in broilers. Indeed, the latter finding has been extensively reported. It is thus not clear why author included inorganic Se with different doses.

In statistics, many interactions on the variables were found. However, no single discussion was stated in the text. Mainly, main factors were discussed.

Specific comments can be found below. 

L2: Title should be corrected as it does not reflect the study. Nano selenium is not main topic either.

L123: 5 replicates per treatment seemed to be low based on the previous studies.

L125: No explanation or basis was given why three inclusion levels per Se sources was used.

L137-138: Zinc analysis?

Table 1: What is toxonil?

Table 1: Nutrient contents expressed percent or g/kg basis.

L98: add “and” between SS and SM

L98: were

L151: crude protein intake was not listed in Table and CP was not measured, but calculated.

L152: PER is not relevant in this study. It is indicator of protein quality.

L152: RGR is not common parameter.

L163: Subheading needs to be changed. Selenium assay in serum samples can be stated in tissue selenium assay section.

L164: trial

L216: tert-butyl hydroperoxide (tert-BHP) was intraperitoneally inject. How sure it produced oxidative stress in this study?

L218: Were tert-BHP injected birds sampled at 40 days?

L238: Tables should be re-arranged. In addition to each treatment, main factors need to be added. This is factorial experiment and there are many interactions on the variables. It needs to be discussed in the text. No discussion on the interaction was given.

L239: PER should be deleted from Result and Discussion.

L246: SeS group exhibited quadratic increase in performance.

L250: compared to SeS?

L270: 0.2mg/kg

L276: Expression “increased protein related to protein digestibility” is not clear.

Table 2. Linear and quadratic p value per selenium source are not needed. If this is needed, then all Tables should be applied.

L303-304: This sentence is not clear.

L327: ALK?

L362: tert-BHP was used to induce oxidative stress in this study. It needs to be discussed here. No control was included here with tert-BHP injection.

L368-370: It is not clear. Rephrase the sentence.

L381: SM?

L389-392: Delete it. Rabbit trial is not relevant.
L439-440: Rephrase the sentence.

L453: Antioxidant activity was determined using three different methodologies. No discussion was given why three methods were used. Results in general increased as selenium levels in diets increased. I am not sure whether increased values indicate higher antioxidant activities based on the methods. Subheadings are not needed as three methods measured oxidative activities. In this section, authors need to present the results and discuss them together.

L498: mg/kg

L504-506: This sentence is not clear.

Author Response

Dear: Dr Reviewer 2

All your comments were taken carefully and corrected in the manuscript

Specific comments can be found below. 

L2: Title should be corrected as it does not reflect the study. Nano selenium is not main topic either.

Response: We accept the reviewer comment and the new titled is:

Dietary modulation of selenium form and level changes its effect on performance, tissue retention, quality of frozen stored meat, and gene expression of antioxidant status in Ross broiler chickens

L123: 5 replicates per treatment seemed to be low based on the previous studies.

Response:

As we are in developing country, our fund for research is limited and we fund it dependently.and the area of work is limited for 5 replicate per treatment,and to compensate thisshortagewe take large number of samples

L125: No explanation or basis was given why three inclusion levels per Se sources was used.

Response: we used three inclusion levels from different Se sourcesat levels rangingaround Se requirements of ross broiler to cover their possible effects.

L137-138: Zinc analysis?

Response: it was corrected to selenium

Table 1: What is toxonil?

Response: toxonil is anti-mycotoxin, we cleared it in table 1.

Table 1: Nutrient contents expressed percent or g/kg basis.

Response: Nutrient contents expressed as percent and this was cleared in table 1 except of metabolizable energy it was per kg of diet and Se mg/kg.

L98: add “and” between SS and SM

Response: It was corrected

L98: were

Response: It was corrected

L151: crude protein intake was not listed in Table and CP was not measured,  but calculated.

Response: this part removed from manuscript

L152: PER is not relevant in this study. It is indicator of protein quality.

Response: We removed it from the manuscript

L152: RGR is not common parameter.

Response: We included it as one of growth parameters. RGR is a measure used to quantify the speed ofanimalgrowth

L163: Subheading needs to be changed. Selenium assay in serum samples can be stated in tissue selenium assay section.

Response: This part was corrected

L164: trial

Response: corrected and removed

L216: tert-butyl hydroperoxide (tert-BHP) was intraperitoneally inject. How sure it produced oxidative stress in this study?

Response:

Tert-butyl hydroperoxide (tert-BHP) was used as prooxidant according to previous published reseaches .Wecollected samples from three birds of each group before injection with (tert-BHP), to study their effect on antioxidant system of the birds (without exposure to oxidative stress), and this data not included in our manuscript as there was no significance between groups (discriped in figure A,C and E). So to confirm our results we select another three birds per group, marked and injected with tert-butyl hydroperoxide, 0.2 mmol/kg body weight, intraperitoneally to induce the oxidative stress.(discriped in figure B,D and F)

This figure showed the difference in birds responses before and after oxidative stress( not included in manuscript)

L218: Were tert-BHP injected birds sampled at 40 days?

Response: This part was addressed.

L238: Tables should be re-arranged. In addition to each treatment, main factors need to be added. This is factorial experiment and there are many interactions on the variables. It needs to be discussed in the text. No discussion on the interaction was given.

Response: Tables rearranged.

Thanks for your comment, as the aim of this study is to compare between different selenium sources and levels, so the main effect of different selenium sources and levels and the interaction between levels and sources is now cleared in all tables and there is no need to add the main factor of each treatment.

In results and discussion: The interaction between selenium sources and levels on different parameters is clear.

L239: PER should be deleted from Result and Discussion.

Response: we deleted it

L246: SeS group exhibited quadratic increase in performance.

Response: Linear and quadratic response was removed as suggested

L250: compared to SeS?

Response: It was added

L270: 0.2mg/kg

Response: It was corrected.

L276: Expression “increased protein related to protein digestibility” is not clear.

Response: It was cleared.

Table 2. Linear and quadratic p value per selenium source are not needed. If this is needed, then all Tables should be applied.

Response: Linear and quadratic p value per selenium source were removed

L303-304: This sentence is not clear.

Response: It was rephrased and cleared

L327: ALK?

 Response: It was deleted.

L362: tert-BHP was used to induce oxidative stress in this study. It needs to be discussed here. No control was included here with tert-BHP injection intraperitonial.

Response: as we mentioned Tert-butyl hydroperoxide (tert-BHP) was used as prooxidant according to previous published reseaches .Wecollected samples from three birds of each group before injection with (tert-BHP), to study their effect on antioxidant system of the birds (without exposure to oxidative stress), and this data not included in our manuscript as there was no significance between groups

L368-370: It is not clear. Rephrase the sentence.

Response: It was rephrased.

L381: SM?

Response: It was cleared.

L389-392: Delete it. Rabbit trial is not relevant.

Response: It was deleted.

L439-440: Rephrase the sentence.

Response: rephrased

L453: Antioxidant activity was determined using three different methodologies. No discussion was given why three methods were used. Results in general increased as selenium levels in diets increased. I am not sure whether increased values indicate higher antioxidant activities based on the methods. Subheadings are not needed as three methods measured oxidative activities. In this section, authors need to present the results and discuss them together.

Response :

We measured the effect of selenium sources and levels on the antioxidant status in tissues and blood to ensure their retention in the body, after that we studied their effect on broiler’s antioxidant system based on gene expression of antioxidant enzymes and antioxidant assay of stored meat in order to justify all possible antioxidant effect of selenium in the body and ensure that selenium will reach to human consumer, thus improving meat quality.

L498: mg/kg

Response: It was corrected.

L504-506: This sentence is not clear.

Response: It was cleared 

Reviewer 3 Report

In this manuscript entitled “Consequences of substituting dietary nano selenium for its other sources on growth, biomarkers of antioxidant capacity of frozen stored meat, and gene expression of antioxidant status in broilers” by Doaa Ibrahim and co-authors presents the results of their studies on fortification of different forms of Se (Met-Se and Nano-Se). The presented data are sound.

There are a few points that have to be addressed and rewrite the manuscripts before publishing: Major revision.

Simple summary:

1.     Line 21: Either change the word “important” to “importance” or remove ‘of’ between important and usage (check grammar).

Abstract:

2.     Line 39: Remove from and line 40: increased muscular PH 24 hours after slaughter should be changed to increased muscular PH after 24 hours of slaughter.

3.     Authors concluded that dietary supplementation of Met-Se and Nano-Se produced acceptable Se levels in chicken meat offered for consumers. I am afraid on what basis authors concluded above statement. It was their assumption without organoleptic tests and human experiment.

4.     What does the study implies or suggests?! Which form of Se is better? Is it Met-Se or Nano-Se?

Materials and methods:

5.     Line 109: Reference should be removed Srivastava and Mukhopaedhyay, 2015 and numbering should be given as per Journal’s instruction.

6.     Section 2.2: Authors are suggested to provide animal ethics approval number.

7.     Line 135: Remove “of the” and include a recent reference for proximate analysis.

8.     Tables are not in the journal’s format. Table 1. Authors are suggested to include brackets for % symbol.

9.     Line 137 to 139: What is meant by Lanthanum 185.4L 50000 mg/Kg? Authors are suggested to covert 50000 mg to grams.

10.  Section 2.3: Authors are suggested to include A K. Panda et al., 2011. Nutritional evaluation and utilization of quality protein maize, Nithyashree maize, and normal maize in broiler chicken. British Poultry Science, 52:5, 632-638.

11.  Section 2.4.3.2: Line 188: Reference number 23 is old. Authors are suggested to include a recent reference. I would suggest to include Lavanya et al., 2012. Evidence-based complementary and alternative medicine. Doi:10.1155/2012/535479.

12.  Line 203: Authors are suggested to include Manjunath et al., 2011. Asian Journal of Experimental Science, vol.25, No.1, 73-80 instead of reference No.25.

Results and discussion:

13.  Section 3.6: Authors are suggested to include a reference at line 436. Lavanya Goodla et al., 2019. Protective effects of Ammannia baccifera against CCl4-induced oxidative stress in rats. 16(8), 1440. Int. J. Environ. Res. Public Health.

14.  All Tables in the manuscript should be represented clearly. Tables are very messy and confusing.

15.  Figure 1.No indication of A and B in the figure 1.

Figure 2. Should be redrawn. It is not very clearly represented. It doesn’t have clear legend explanation. Figure 2B. Is not identified and authors should indicate which figure is for which antioxidant enzyme gene expression levels.

16.  Statistical representation in the footnotes of Tables and Figures are not clearly represented.

17.  Authors are advised to reduce the number of reference and also take care to include most recent references.

18.  How authors have decided that supplementation of Met-Se and Nano-Se improved the growth, performance and antioxidant levels in chicken without employing a positive control compound which is available in market. Could you include positive control? Why/why not?

19.   Authors are suggested to strengthen the importance and relevance of the present results in the discussion section.

20.  Authors should check the conclusion part: the results are really supporting the conclusion? The tested /selected parameters are sufficient for the conclusion?

21.  References should be cited by following journal style/format.

22.   Need to check for typographical errors, plagiarism, punctuation, and grammar throughout the manuscript.

Author Response

Dear: Dr Reviewer 3 

all your comments were taken seriously and as possible as it was corrected 

Simple summary:

1.      Line 21: Either change the word “important” to “importance” or remove ‘of’ between important and usage (check grammar).

Response: It corrected.

Abstract:

2.      Line 39: Remove from and line 40: increased muscular PH 24 hours after slaughter should be changed to increased muscular PH after 24 hours of slaughter.

Response: It corrected.

3.      Authors concluded that dietary supplementation of Met-Se and Nano-Se produced acceptable Se levels in chicken meat offered for consumers. I am afraid on what basis authors concluded above statement. It was their assumption without organoleptic tests and human experiment.

Response:

In our study we measured selenium content in blood, liver and meat (breast meat), the results revealed that supplementation of Met-Se and Nano-Se in broiler’s diet, increased retention of Se in breast meat with increasing their level when compared to supplementation of SeS.  The safe limit of Se in human food has been established at 2.0 mg/kg for the United States [AAFCO. Official guidelines suggested for contaminants in individual mineral feed ingredients. Official publication. Association of American Feed Control Officials Inc., Olympia, WA 2011., 304]. This level agreed with our results that up to 0.6 mg/kg of Se in broiler diets precipitates less than 1 mg/kg in meat with all sources of Se and this was sufficient for se requirement for human. Moreover, the level of Se in meat was increased by Met-Se and Nano-Se supplementation.

What does the study implies or suggests?! Which form of Se is better? Is it Met-Se or Nano-Se?

Response:

Our study suggested that Nano-Se was more potent than Met-Se, which in turn was more potent than inorganic Se against oxidative stress, which improved the quality of meat under frozen conditions. Dietary supplementation of either Met-Se or Nano-Se up to 0.6 mg/kg increased broiler’s performance and was more efficiently retained in the body than SeS and consequently the offered dietary levels of Se available for human consumer.

Materials and methods:

4.      5.    Line 109: Reference should be removed Srivastava and Mukhopaedhyay, 2015 and numbering should be given as per Journal’s instruction.

5.      Response: Referencerevised and corrected

6.     Section 2.2: Authors are suggested to provide animal ethics approval number.

 Response: It was inserted. (Approval no: ZU-IACUC/2/F/123/2018)

7.     Line 135: Remove “of the” and include a recent reference for proximate analysis.

Response: It was corrected

8.     Tables are not in the journal’s format. Table 1. Authors are suggested to include brackets for % symbol.

Response: It formatted and reorderd

9.     Line 137 to 139: What is meant by Lanthanum 185.4L 50000 mg/Kg? Authors are suggested to covert 50000 mg to grams.

Response: It converted to gram

10.  Section 2.3: Authors are suggested to include A K. Panda et al., 2011. Nutritional evaluation and utilization of quality protein maize, Nithyashree maize, and normal maize in broiler chicken. British Poultry Science, 52:5, 632-638.

11. Section 2.4.3.2: Line 188: Reference number 23 is old. Authors are suggested to include a recent reference. I would suggest to include Lavanya et al., 2012. Evidence-based complementary and alternative medicine. Doi:10.1155/2012/535479.

Response: Thereferences was added

12.  Line 203: Authors are suggested to include Manjunath et al., 2011. Asian Journal of Experimental Science, vol.25, No.1, 73-80 instead of reference No.25.

Response: The references was added

Results and discussion:

13.  Section 3.6: Authors are suggested to include a reference at line 436. Lavanya Goodla et al., 2019. Protective effects of Ammannia baccifera against CCl4-induced oxidative stress in rats. 16(8), 1440. Int. J. Environ. Res. Public Health.

Response: The references was added

14.  All Tables in the manuscript should be represented clearly. Tables are very messy and confusing.

Response: Tables was reformatted and orderd

15.  Figure 1.No indication of A and B in the figure 1.

Response: It was addressed

Figure 2. Should be redrawn. It is not very clearly represented. It doesn’t have clear legend explanation. Figure 2B. Is not identified and authors should indicate which figure is for which antioxidant enzyme gene expression levels.

Response: It was addressed

16.  Statistical representation in the footnotes of Tables and Figures are not clearly represented.

Response: It was addressed and cleared

17.  Authors are advised to reduce the number of reference and also take care to include most recent references.

Response: Some references wereremoved and other was updated and then some new references was added according to the suggestions of the reviewers. As our manuscript contained different sources and levels of selenium. There were a lot of results in our study, so all results should be supported by references.

18.  How authors have decided that supplementation of Met-Se and Nano-Se improved the growth, performance and antioxidant levels in chicken without employing a positive control compound which is available in market. Could you include positive control? Why/why not?

Response: In our experiment, we used the inorganic form of selenium which is available in market and commonly used in feed premix (vitamins and minerals mixtures), so we compare between the commercial inorganic form of selenium (selenium selenite) and others sources including organic and nano form of selenium.

19.   Authors are suggested to strengthen the importance and relevance of the present results in the discussion section.

Response: Our manuscript described in details to which extent organic and nano selenium form affecting on selenium bioavailability and  broiler’s antioxidant system when compared with a commercially available form of inorganic selenium, this was clear in our results  and discussion which supported by many authors. Moreover, it is the first time to make such meat antioxidant assay in ross broiler included in our manuscript. So, we think that our manuscript support previous studies and add new information about how organic and Nano selenium will affect meat quality.

20.  Authors should check the conclusion part: the results are really supporting the conclusion? The tested /selected parameters are sufficient for the conclusion?

Response: We think that our conclusion was concentrated and described that using of Met-Se or Nano-Se up to 0.6 mg/kg increased broiler performance based on our results. Also, Nano-Se was more potent than Met-Se, which in turn was more potent than inorganic Se against oxidative stress, which improved the quality of meat under frozen conditions (in this section we measured the antioxidant effect of selenium sources and levels in tissues and blood to ensure their retention in the body, after that we studied their effect on broiler’s antioxidant system based on gene expression of antioxidant enzymes and antioxidant assay of stored meat  in order to justify all possible antioxidant effect of selenium in the body and ensure that selenium will reach to human consumer through meat, thus improving meat quality.

21.  References should be cited by following journal style/format.

Response: It was corrected.

22.   Need to check for typographical errors, plagiarism, punctuation, and grammar throughout the manuscript.

We checked all of these as possible and w e send language editing certificate to the editor

Round 2

Reviewer 2 Report

In the first review, reviewer raised the concern three major points. In the revised manuscript, authors did not response these points. First of all, recent paper published (in brit pout sci) already with similar experimental setting reported the functions of selenium broiler chickens. Authors did not respond this concern. Secondly, in the introduction, organic selenium is preferable to inorganic selenium in broilers. It is still not clear why authors used inorganic selenium. If authors did not use it, then replication numbers could be increased. Finally, almost all variables measured exhibited interaction. Then, main factors should not be discussed, which were not the case in this manuscript. I suggest authors to consider the above three points in revising manuscript.

Author Response

Dear Dr Reviewer 2

thank you for your valuable comments in revising our manuscript we had addressed all comments and we hope to accept our responses 

First point: recent paper published (in brit pout sci) already with similar experimental setting reported the functions of selenium broiler chickens.

Despite of the authors of the aforementioned article used different sources of selenium as our paper, there are many viable differences between this paper and our paper.

-          First of all, the size of NS particle in this paper was ± 60 nm and in our manuscript was 42 nm. This difference in size of nano particles affected selenium bioavailability and retention, consequently on selenium antioxidant function that was evaluated and reported via various advanced ways than in suggested paper and brought the second difference which is the quantification of antioxidants genes by real time PCR which is more accurate and sensitive to changes of different levels of selenium and nano particles size and this point not analyzed in suggested paper

-          Third and a novel difference in our manuscript which has not yet been applied in ross broiler meat is measuring the total antioxidant capacity of meat using precise assays as DPPH Assay, ABTS and FRAP assays.

-          Using three different levels from selenium sources around ross broiler requirement will give more accurate choice to the best level suggested to be generalized in poultry farms than using two levels from selenium.

-          According to these above differences we suggested that our manuscript added new data and provides a broad-spectrum analysis based on more accurate and advanced techniques to our Knowledge.

Secondly, in the introduction, organic selenium is preferable to inorganic selenium in broilers. It is still not clear why authors used inorganic selenium. If authors did not use it, then replication numbers could be increased.

Even though organic selenium is clearly preferable than inorganic selenium in broilers, the use of inorganic selenium forms as selenium selenite is still a common practice used in several commercial premixes until now. Thus, comparing the other forms and sources to a common source should point out clearly the difference between inorganic selenium (commonly used) and other recent sources suggested to be added in broiler ration to poultry farmers. Eventually, poultry farmers can decide on their own whether they will replace inorganic sources of selenium by other new sources of selenium or not. Collectively, we think that presence of inorganic selenium is inevitable in any comparative study regarding selenium sources including our manuscript.

Finally, almost all variables measured exhibited interaction. Then, main factors should not be discussed, which were not the case in this manuscript.

To the best of our knowledge, we have removed the main effect from our manuscript and addressed the interactions between sources and levels of Se as described in the track changes. For further modifications, we would appreciate if there are further changes please specify them.

Reviewer 3 Report

Authors have now improved the manuscript, still few points should be taken care off, including

Figure 1 legend: Line 115 'a' should be capitalized (A) and 'b' also should be capitalized (B).

Tables 2 and 3: not clear and fussy again. should be reordered.

Figure 2: Line 361 include (A) before glutathione peroxidase, (B) before superoxide dismutase, and (C) before catalase.

Once again check for plagiarism, spelling and typographical errors through out the manuscript.

Author Response

Dear Dr Reviewer 3

thank you for revising our manuscript and we responded to all your comments as  

Figure 1 legend: Line 115 'a' should be capitalized (A) and 'b' also should be capitalized (B).

Response:  corrected

Tables 2 and 3: not clear and fussy again. should be reordered.

Response:arranged and cleared as this may be due to track changes correction and  after accepting this changes it is arranged

Figure 2: Line 361 include (A) before glutathione peroxidase, (B) before superoxide dismutase, and (C) before catalase.

Response:addressed

Once again check for plagiarism, spelling and typographical errors throughout the manuscript.

Response: addressed

Round 3

Reviewer 2 Report

Authors well addressed the comments raised by the reviewer.

The following recommendations can be addressed during the proof reading.

I do not understand why authors did not cite the recommended paper which had same topic and similar experimental settings although there were visible (?) differences. The reviewer think that the paper was the most relevant to the current experiment. I suggest authors to cite the paper recommended int he Introduction and/or Discussion for comparison.

Few editorial suggestions are as follows:

The following sentences that authors used “interaction” are not logically correct. I have simplified the sentences for authors’ review.

L236: the interaction …….. had significant effect.

L240: the interaction …. plays an important role…

L279: the interaction between . ……. affected Se concentration…

L326: the expression pattern ….. in relation to the interaction between …. was varied.

Author Response

Dear Reviewers   

Thank you for your effort in our manuscript the missing points have been addressed 

Second reviewers:

I do not understand why authors did not cite the recommended paper which had same topic and similar experimental settings although there were visible (?) differences. The reviewer think that the paper was the most relevant to the current experiment. I suggest authors to cite the paper recommended int he Introduction and/or Discussion for comparison.

Response:  the reference included in discussion

The following sentences that authors used “interaction” are not logically correct. I have simplified the sentences for authors’ review.

L236: the interaction …….. had significant effect.

Response: corrected

L240: the interaction …. plays an important role…

Response: corrected

L279: the interaction between . ……. affected Se concentration…

Response: corrected

L326: the expression pattern ….. in relation to the interaction between …. was varied.

Response: corrected
